# Hyperbaric Oxygen Therapy in Systemic Inflammatory Response Syndrome

**DOI:** 10.3390/vetsci9020033

**Published:** 2022-01-18

**Authors:** Débora Gouveia, Mariana Chichorro, Ana Cardoso, Carla Carvalho, Cátia Silva, Tiago Coelho, Isabel Dias, António Ferreira, Ângela Martins

**Affiliations:** 1Arrábida Veterinary Hospital—Lisbon Animal Regenerative and Rehabilitation Center, 2675-655 Odivelas, Portugal; anacardosocatarina@gmail.com (A.C.); mv.carla.c@gmail.com (C.C.); catiamsilva@outlook.com (C.S.); tiagoccoelho@netcabo.pt (T.C.); vetarrabida.lda@gmail.com (Â.M.); 2Superior School of Health, Protection and Animal Welfare, Polytechnic Institute of Lusophony, Campo Grande, 1950-396 Lisboa, Portugal; 3School of Agrarian and Veterinary Sciences, Department of Veterinary Science, University of Trás-os-Montes and Alto Douro, 5000-801 Vila Real, Portugal; mariana.chfa@gmail.com (M.C.); idias@utad.pt (I.D.); 4Faculty of Veterinary Medicine, University of Lisbon, 1300-477 Lisbon, Portugal; aferreira@fmv.ulisboa.pt; 5CIISA—Centro Interdisciplinar-Investigação em Saúde Animal, Faculdade de Medicina Veterinária, Av. Universidade Técnica de Lisboa, 1300-477 Lisbon, Portugal; 6Faculty of Veterinary Medicine, Lusófona University, Campo Grande 376, 1749-024 Lisbon, Portugal

**Keywords:** SIRS, HBOT, traumatic, dogs, ATA

## Abstract

(1) Background: Systemic inflammatory response syndrome (SIRS) can occur due to a large number of traumatic or non-traumatic diseases. Hyperbaric oxygen therapy (HBOT) may be used as a main or adjuvant treatment for inflammation, leading to the main aim of this study, which was to verify the applicability of HBOT as a safe and tolerable tool in SIRS-positive dogs. (2) Methods: This prospective cohort study included 49 dogs who showed two or more parameters of SIRS, divided into the Traumatic Study Group (n = 32) and the Non-Traumatic Study Group (n = 17). All dogs were submitted to HBOT for 60–90 min sessions, with 2.4–2.8 ATA. (3) Results: This study revealed that 73.5% (36/49) of dogs showed improvement, and the minimum number of HBOT sessions was two, with a mean of 12.73. The number of days between diagnosis and the beginning of HBOT showed statistical significance (*p* = 0.031) relative to the clinical outcome. No dogs showed any major side effects. (4) Conclusions: We concluded that HBOT may be safe and tolerable for SIRS-positive dogs, and that it should be applied as early as possible.

## 1. Introduction

The development of systemic inflammatory response syndrome (SIRS) can be due to a large number of diseases that evoke activation of the inflammatory cascade, such as sepsis, heatstroke, pancreatitis, immunomediated diseases, cancer, severe trauma, or burns [1,2,3].

Ischemia is a frequent complication of traumatic events, and, when reperfusion occurs, an ischemia–reperfusion injury might develop. This complication is caused by the quick release of toxins and free radicals, which were previously restrained by the lack of perfusion in the ischemic area, into the bloodstream. This is an oxidative injury that is further exacerbated by the release of inflammatory mediators, arterial vasoconstriction, thrombosis, and leucocyte adhesion to the endothelium [4,5].

At a cellular level, ischemia may induce an anaerobic metabolism, with consequent diminution of adenosine triphosphate (ATP) production and malfunction of ion channels, which leads to an increase in cellular volume, due to fluids entering the cell, and compromises cytoplasmic enzymatic activity. After prolonged ischemia, metabolic acidosis can also occur, due to the metabolic residues withheld in the cells.

In the reperfusion phase, mitochondrial lesions and electrolyte imbalance contribute to the occurrence of oxidative stress by the nicotinamide adenine dinucleotide phosphate (NADPH) oxidase, nitric oxide (NO) synthase, and xanthine oxidase systems. The retention of reactive oxygen species (ROS) leads to cellular lesions and, later, to cellular death through autophagia, apoptosis, mitoptosis, necrosis, or necroptosis [4,6].

Hyperbaric oxygen therapy (HBOT) is based on the administration of approximately 100% oxygen inside a chamber pressurized to more than 1 Atmospheres Absolute (ATA). This therapy is used as a main or adjuvant treatment for inflammation, extensive and/or chronic wounds, ischemia, and infections [6,7]. 

To maintain normal cellular metabolism, tissues require a mean of 60 mL of oxygen per liter of blood, in the presence of normal perfusion. At 1 ATA (equivalent to sea level), the oxygen concentration in the blood is around 3 mL/L. Thus, when the pressure rises to 3 ATA, the oxygen dissolved in the plasma reaches a concentration of 60 mL/L, thereby allowing much more efficient oxygenation, since the minimum oxygen required to sustain the tissues is met without the need for oxygen linked to hemoglobin [8]. The minimum pressure required for therapeutic effects of HBOT is 1.4 ATA, and the maximum pressure recommended is 3 ATA, at which point HBOT may not be safe or tolerable.

The physiology of HBOT is complex; however, multiple studies have demonstrated the main effects of HBOT on inflammatory and healing mediators (Table 1).

We hypothesized that HBOT may be a useful adjuvant therapy for SIRS-positive dogs, of traumatic or non-traumatic origin. Thus, the aim of our study was to assess the safety, tolerability, and applicability of HBOT in SIRS-positive dogs, due to traumatic or non-traumatic causes.

## 2. Materials and Methods

This prospective cohort study was conducted at the Lisbon Animal Rehabilitation and Regeneration Center between 20 August 2020, and 20 February 2021. This study addressed dogs with SIRS of traumatic or non-traumatic etiology.

The clinical study followed the guidelines proposed by Memar and collaborators (2019) [7] regarding the pressure used in each HBOT treatment, ranging from 1.4 to 3 ATA. This study population focused on SIRS positive dogs, and included 49 dogs who showed two or more parameters of SIRS [1] (Table 2), regardless of etiology, sex, age, breed, or weight.

The study population was divided into two groups, concerning the etiology of SIRS: the Traumatic Study Group (TSG) (represented by 32 dogs) and the Non-Traumatic Study Group (NTSG) (represented by 17 dogs). Exclusion criteria included contraindications to HBOT, such as superior or inferior respiratory infections, for example, hemothorax/pneumothorax in the TSG or pulmonary fibrotic syndromes in the NTSG, as well as chronic stenotic otitis and/or tympanic membrane perforations (in NTSG dogs).

The clinical study consisted of 29 dogs of known breed—Labrador Retriever was the most prevalent (6 dogs), followed by German Shepherd (5 dogs)—and 20 mixed breed dogs. Of these 49 dogs, 27 were males and 22 were females. The mean age of the studied population was 8.85 years (minimum 4 months and maximum 19 years), and the mean weight was 23.22 kg (minimum 1 kg and maximum 58 kg).

The TSG was represented by 24 dogs with inflammatory, necrotic, and/or infected tissue discontinuity trauma, and 8 dogs with central nervous system trauma, with or without bone involvement.

The NTSG was represented by 5 dogs with anemia and 12 dogs without anemia, of which 8 had a metabolic disease and 4 had a central nervous system degenerative disease.

### 2.1. Admission Consultation

To be admitted to the study, all dogs underwent a consultation, performed by a veterinary surgeon certified by the Hyperbaric Veterinary Medicine and International ATMO^®^ (International ATMO, San Antonio, TX, USA), as safety director.

At the admission consultation, a physical examination and assessment of vital parameters (heart rate, respiratory rate, perfusion parameters and rectal temperature) were performed, and any deviation from normal ranges were recorded.

In addition, blood analyses were conducted, radiographic and ultrasound images were taken, and, finally, the ear canal was properly and rigorously evaluated, as well as the tympanic membrane.

During the admission consultation, all dogs were classified according to the previously mentioned SIRS parameters. The characterization of the sample and admission to the study are represented in Figure 1 and Table 3.

### 2.2. HBOT Protocol

The hyperbaric chamber used in this study was a type C chamber, meaning a monoplace chamber (HVM^®^), exclusively for veterinary medicine.

The protocol consisted of seven phases:Physical examination of the animal, verifying the absence of counterindications for HBOT, and, later, removal of bandages and collars that are not advised during HBOT. IV catheters, 100% protected by cotton bandages, were also placed.Switching on the hyperbaric chamber, testing the ground connection (demonstrating values between 0 and 1 Ω) to ensure safe use and opening of the O_2_ tap, after verification of the O_2_ levels in the cylinders.Placement of the dog inside the chamber, after humidifying its skin with a sprayer or a wet cloth, and closure of the door, and making sure there are no air leakages (Figure 2).Compression phase: opening of the O_2_ valve and increase of the pressure to 5 psi, followed by a 2-psi increase per minute, until a maximum of 30 psi, or until the animal shows signs of discomfort, to correct major or minor secondary effects that may occur.Treatment phase: maintenance of the pressure between 2.4 and 2.8 ATA for 30–45 min, except when minor secondary effects are shown.Decompression phase: closure of the O_2_ valve and slow decompression of the chamber. When the pressure is close to 0 psi, two cranks are slightly opened to let out some of the air, followed by the total opening of the door.Cleaning of the interior of the chamber with 50 mL of 4% chlorhexidine solution diluted in 1 L of NaCl. Closure of the safety oxygen tap, as well as the pressure regulator and the flux sensor pressure gauge.

The number of sessions was decided according to the needs of each animal, whilst respecting the checklist of minor and major side effects. A rule of at least two treatments per animal was implemented. The first session had a longer treatment phase, and the HBOT sessions were executed up to two times a day, although animals did not receive HBOT every day.

The total duration of one HBOT session could last 60 to 90 min, considering that 15 to 20 min is the compression phase, and 20 to 25 min is the decompression phase.

### 2.3. Internal Medicine Protocol

In the present study, all animals in the TSG and NTSG received hemodynamic stabilization procedures, according to individual vital parameters and analytic values.

The antibiotic therapy protocol used in dogs with clinical signs of infection, which was 73.5% (36/49) of dogs, was an association between beta-lactam antibiotics (ampicillin 10 mg/kg IV TID), quinolones (enrofloxacin 5 mg/kg IV SID), and metronidazole (12.5 mg/kg IV BID), when there was no bone involvement. When bone infection was present, clindamycin (11 mg/kg PO BID) was administered.

All dogs with signs of infection were subjected to culture analysis and sensibility tests at admission and each 8–10 days during treatment, in order to change the standard antibiotic protocol, if needed. 

### 2.4. Monitoring of Side Effects

During the HBOT procedure, all side effects were observed, and major and minor side effects were registered, as shown in Table 4.

### 2.5. Study Population Evaluation

All TSG dogs were monitored daily following the SIRS scale every 2, 4, 6, 8, or 12 h, according to their hemodynamic stability. For all patients that showed a predisposition for sepsis, the quick Sepsis Related Organ Failure Assessment (qSOFA) scale was calculated, assessing the serum lactate levels and systolic arterial pressure. For some of the patients with central nervous system trauma, the modified Glasgow scale was used, valuing the mental state of the animals. To evaluate the NTSG dogs, the SIRS scale was used.

In both groups, analytic complementary exams were performed, such as hemograms and creatinine, urea, glucose, albumin, total proteins, alkaline phosphatase, alanine transaminase, gamma-glutamyl transferase, and total bilirubin levels.

### 2.6. Statistical Analysis

The software used to create the database and for the statistical analysis were the Microsoft Office Excel 365^®^ (Microsoft Corporation, Redmond, WA, USA) datasheet and the statistical analysis software IBM SPSS Statistics 25^®^ (International Business Machines Corporation, Armonk, NY, USA).

The descriptive statistical analysis allowed the sample characterization regarding the frequencies of the studied variables. The continuous quantitative variables studied were: weight, age, number of HBOT sessions, number of days between the first and the last HBOT sessions and number of days between the diagnosis and the first HBOT session. The qualitative variables were: clinical outcome, race, sex, and etiology.

Two discrete quantitative variables (lactate serum levels and systolic arterial pressure) and one ordinal qualitative variable (modified Glasgow scale) were also evaluated.

In the inferential statistical analysis, the normalcy of the study data was verified with the Shapiro–Wilk test, resulting in *p* > 0.05.

In this study, the parametric *t*-test for independent samples and the non-parametric Chi-square and Mann–Whitney tests were performed (Table 5), to evaluate the relations between relevant variables, namely:Etiology and number of HBOT sessions;Etiology and number of days between the first and the last HBOT sessions;Etiology and number of days between the diagnosis and the first HBOT session;Etiology and clinical outcome.

For both the parametric and non-parametric tests, *p* ≤ 0.05 was considered the level of statistical significance.

## 3. Results

In this study, considering the total sample (n = 49), a normal distribution regarding age and weight was obtained, demonstrated by their respective histograms. To prove this normalcy, with n < 50, the Shapiro–Wilk test was performed, confirming the results regarding the *age* category (*p* = 0.389). However, in the *weight* category, a *p* value of 0.046 was obtained, indicating a borderline normality.

As previously explained, the study sample was divided between the TSG and NTSG groups and, when the Shapiro–Wilk test was applied, a *p* value of >0.05 was obtained in both groups and both categories.

In the NTSG, there were no signs of possible sepsis, with negative qSOFA and normal lactate levels, systolic arterial pressure, and modified Glasgow scale values. The following table explains the evolution in the TSG (Table 6). All dogs of both groups had normal values at the time of the first HBOT. 

Regarding the *etiology*, 65.3% (32/49) of dogs presented a traumatic etiology and 34.7% (17/49) presented a non-traumatic etiology.

As for the number of days between the diagnosis and the first HBOT session, we observed that 51% (25/49) of the dogs started HBOT in 7 days or less after diagnosis, 32.7% (16/49) between 8 and 15 days after diagnosis, and 16.3% (8/49) more than 15 days after diagnosis.

In the TSG, 56.3% (18/32) started treatment in 7 days or less after diagnosis, 31.3% (10/32) between 8 and 15 days and 12.5% (4/32) more than 15 days after diagnosis. In the NTSG, 41.2% (7/17) started treatment in 7 days or less after diagnosis, 35.3% (6/17) between 8 and 15 days and 23.5% (4/17) more than 15 days after diagnosis.

The minimum number of HBOT sessions in an animal recorded in this study was two and the maximum was 100, with a mean of 12.73 (SD ± 15.19 days).

This variable was categorized, and the results were that 44.9% (22/49) of dogs underwent five or less HBOT sessions, 12.2% (6/49) between 6 and 10 HBOT sessions, and 42.9% (21/49) more than 10 HBOT sessions. Fifty percent (16/32) of the TSG dogs underwent five or less sessions, while 52.9% (9/17) of the NTSG underwent more than 10 sessions, considering these as a possible information bias.

Regarding the number of days between the first and the last HBOT sessions, a minimum of 3 and a maximum of 496 were recorded, with a mean of 63.78 (SD ± 102.36 days); 42.9% (21/49) of patients remained in treatment for 15 days or less, 32.7% (16/49) for more than 50 days, and 24.5% (12/49) between 16 and 50 days. Among the TSG dogs, 46.9% (15/32) remained in treatment for a maximum of 15 days, and as for the NTSG, 41.2% (7/17) remained in treatment for more than 50 days. The number of days under HBOT in relation to the etiology showed *p* = 0.626. The total number of HBOT sessions during the study was 624, with 398 belonging to TSG patients and 226 belonging to the NTSG dogs.

Regarding the clinical outcome, 73.5% (36/49) of dogs showed improvement, 24.5% (12/49) were euthanized or died, and 2% (1/49) presented no changes in outcome. When comparing the TSG to the NTSG patients, regarding the clinical outcome, no statistical significance was obtained (*p* = 0.360).

In the total sample, the number of treatments and the number of days between the first and the last HBOT sessions did not show statistical significance regarding the clinical outcome (*p* = 0.752 and *p* = 0.331, respectively).

As for the number of days between the diagnosis and the beginning of HBOT, a statistical significance was shown (*p* = 0.031) relative to the clinical outcome.

In regard to the side effects, no dog showed any of the major side effects, in either the TSG or TNSG. The minor side effects experienced are represented in Table 7, with head shaking presented by 100% of dogs in both groups.

## 4. Discussion

In this study, a populational sample of n = 49 dogs was studied, according to the cohort guidelines. The distribution of the sample presented normalcy regarding the age and weight categories, as the mode, median, and mean were similar.

As previously reported, the Shapiro–Wilk test confirmed normalcy regarding the age category. However, in the weight category, the mean was 23.22 kg, while the median and mode were 25 kg; therefore, it was classified as borderline normalcy.

Since the Shapiro–Wilk test showed a *p* value of >0.05, a comparison of the TSG and NTSG was possible. In clinical practice, it is important to know how traumatic and non-traumatic etiologies vary with HBOT, which justifies this division of the population sample.

The study showed that, in both groups, HBOT was safe and tolerated, as not one patient needed emergency protocols in the decompression phase, and nor were there major side effects registered, such as barotrauma, seizures, syncope, or death; these results are in accordance with Gouveia et al. (2021) [10].

The use of this technique as a complementary therapy for SIRS patients was based on the physiopathology of SIRS and the possible role of HBOT in the inflammatory process, as it may decrease proinflammatory cytokine production and the release of tumor necrosis factor (TNF-α), while increasing the production of vascular endothelial growth factor (VEGF) and fibroblast growth factor (FGF), both known for their respective roles in the inflammatory process [11,12].

Furthermore, oxygen has been used in surgical and anemic patients, in patients with chronic or difficult wounds, in the control of infections associated with the presence of implants, and in burns, as described by different authors [13,14,15,16], therefore confirming the efficacy of HBOT as an adjuvant therapy due to its anti-hypoxic and anti-anoxic effects [17], which were essential for 75% (24/32) of the TSG dogs, as they showed an inflammatory, necrotic, and/or infected tissue discontinuity of traumatic etiology, although with normal qSOFA when the first HBOT session was performed.

SIRS can be caused by depletion of antioxidant reserves, the release of inflammatory mediators, arterial vasoconstriction, thrombosis, and endothelium–leucocyte cellular adhesion. It can lead to the reduction of ROS and NO, which aid systemic inflammatory response [5]. All of these phenomena may lead to multiple organ failure, and even death. Therefore, HBOT is used to diffuse intracellular oxygen, promoting neovascularization and post-ischemia tissue oxygenation [5,18], which was essential for the 70.6% (12/17) of the NTSG dogs that did not show hemoglobin or hematocrit changes. Of these, 8 dogs had metabolic disease, such as renal failure.

In mice, a decrease in neutrophils was demonstrated in various tissues, such as the lung, brain, and intestinal mucosa, in response to the inflammatory process, as well as a decrease in serum concentration of TNF-α, which occurs in ischemia–reperfusion injury and stroke [19]. It was also suggested that, in cats, HBOT increases the production of collagen, allowing osteoblastic activity to occur [20]. 

Wu et al. (2021), McCurdy et al. (2021), and Albrethen et al. (2020) [21,22,23] studied the role of HBOT in Crohn’s disease and ulcerative colitis, and demonstrated its adjuvant role in acute (TSG) and chronic (NTSG) SIRS, although more case reports and blind, randomized, controlled studies are needed.

As reported by Al-Waili and Butler (2006) [24], HBOT also promotes NO regulation, essential for the balance of vasoconstriction and vasodilation processes. By promoting the release of VEGF, HBOT increases capillary permeability and endothelial NO synthase, and has antioxidant and endothelial integrity control properties, which are essential for treatment of sepsis or multiorgan failure predisposition situations with tissue ischemia involvement [25]. This was pointed out by Shmalberg et al. (2020) [26], when describing the case of a dog with *Lagenidium giganteum*, which caused diffuse inflammation, vascular thrombosis, and secondary ischemia in several parts of the brain.

In our study, the minor side effects were few. Only head shaking was present in all 49 dogs (100% both groups), most likely due to the selection of dogs with a normal and clean ear tract and tympanic membrane, revealing the importance of the admission consultation, which includes an ear exam that should be performed before and after HBOT. A higher number of minor side effects was registered in the NTSG, as the mental state of the TSG dogs could be altered. However, in both groups, HBOT was revealed to be safe, with no major issues, which was established by the 624 treatments that were performed.

In the TSG, 56.3% (18/32) of the dogs underwent HBOT in 7 or less days after diagnosis, and 41.2% (7/17) of the NTSG started treatment in the same period. This early use of HBOT was likely the reason that 73.5% (36/49) of patients showed a positive outcome (*p =* 0.031).

In human medicine, the early use of HBOT in ischemic strokes [27] and ischemia–reperfusion injuries [25] has been shown to have neuroprotective effects. This was also demonstrated by Marcinkowska et al. (2021) [28], in humans, and Yang et al. (2010) [29] in mice, affirming that the neurologic deficits and neural mass loss are considerably reduced after early HBOT (3 to 12 h after injury, or 24 to 72 h after, when multiple sessions were performed).

Yin and Zhang (2005) [30] demonstrated that repeated treatments are important, and, while there was no difference between three and five continuous treatments, beneficial effects after the first treatment, which should be prolonged, were observed. Baratz-Goldstein et al. (2017) [31] showed, in mice, that for the stimulation of cognitive learning abilities, treatments still have a positive effect even when started later.

Several studies have also used HBOT in focal cerebral ischemia, involving the precocious use of this therapy, considering that the first session should be done as early as possible [27,32,33,34,35,36,37,38,39,40,41,42,43,44]. 

According to the Undersea and Hyperbaric Medical Society, the pressure exerted in HBOT can be 1.4 ATA or higher. However, the indications approved by this entity determine that the HBOT pressure should be at least 2 ATA [45,46]. In this way, the “low-pressure” chambers, which use pressures ranging from 1.2 to 1.3 ATA, are frequently used for plastic issues [19].

In our study, the treatment phase used pressures between 2.4 and 2.8 ATA, not following the ones determined by Birnie et al. (2018) [47], since we wanted the most therapeutic effect possible. The treatments that are frequently used in human medicine, in the resolution of chronic and difficult wounds, are at a pressure of 2.4 ATA, where the seizure incidence reported was 1.3 occurrences in 10,000 treatments. This pressure was also used by Gouveia et al. (2021) [10] in 289 HBOT sessions in veterinary medicine.

In this study, a total of 624 HBOT sessions were performed, without any major side effects, and allowing a plasmatic oxygen concentration of 60 mL/L, while having a larger interaction in ischemic tissues, associated with the potentiation of antibiotic therapeutic effects, mainly quinolones [48]. HBOT also allows for a balance between neutrophils and endothelial cells, which promotes arterial vasoconstriction and, consequently, cellular apoptosis [49].

As reported by Andrade et al. (2016) [50], a pressure of 2.4 ATA is the most frequent. In our study, NTSG patients allowed us to increase the pressure to 2.8 ATA faster in the first treatment, although the first HBOT session was always longer, as previously stated. In the TSG patients, the chamber pressure did not always reach 2.8 ATA.

Thus, our study protocol followed many authors in this regard [21,51,52,53,54,55,56,57,58,59,60,61,62,63,64,65].

In this study, the duration of HBOT treatments was 60 to 90 min, in compliance with Amaral et al. (2021) [66] and McCurdy et al. (2021) [22], who stated that the duration of the treatments is important for clinical success.

In our study, there was no correlation between the number of days until the first HBOT session and the etiology. However, 50% (16/32) of TSG patients underwent five or fewer treatments, while 52.9% (9/17) of NTSG dogs underwent more than 10 HBOT sessions. This could be because the non-traumatic diseases were chronic situations, where HBOT was used as part of a palliative approach.

In human medicine, HBOT is used in 45% of patients with cognitive diseases, having a positive effect in all of these patients [28,67].

The therapeutic neuropsychological rehabilitation programs are more effective when used early on [68,69,70,71,72,73]. However, there are no efficient neuropsychological rehabilitation programs for chronic cases [68,69,70,71,72,73,74].

The mean number of days between the first and the last HBOT sessions of this study was 63.78 (SD ± 102.36 days), with a minimum of 3 and a maximum of 496 days. In the TSG, the maximum days in treatment was 15 days, while in the NTSG, patients underwent treatment for more than 50 days. Thus, HBOT may be applicable in acute situations that require early treatment, but could also be applied as a long-term therapeutic approach. 

This study compared two groups of patients, in which the sample size was reduced, and their division was not blind or randomized, having been divided according to the etiology of the patients. During the study, we felt the need for an approved monitoring scale for the NTSG patients, as the evaluation of these dogs according to the SIRS parameters, associated with analytic complementary exams, did not allow pragmaticism. This is extremely important, as it gave a biased view of the study, which may justify the difference in the number of days between the first and last HBOT sessions between the two studied groups. Although the *p* value did not show significance, caution in the interpretation of these results is needed.

The association of these two limitations does not allow for results without variability, and it confirms the absence of significance between the number of treatments and the clinical outcome, as well as the number of days between the first and the last HBOT session and the clinical outcome. Thus, future studies with larger cohort sizes are needed to further explore this paradigm. 

## 5. Conclusions

We conclude that HBOT is safe and tolerable, as no major side effects were recorded, and there were few minor side effects, in 624 HBOT sessions.

This study showed that the number of days between diagnosis and the first HBOT session affects the clinical outcome, indicating that this therapy should be applied as early as possible.

A positive outcome was observed in 73.5% (36/49) of patients, indicating the efficacy of the therapy. It is also suggested that HBOT is a possible adjuvant therapy for stabilizing SIRS patients with a traumatic or non-traumatic disease. 

## Figures and Tables

**Figure 1 vetsci-09-00033-f001:**
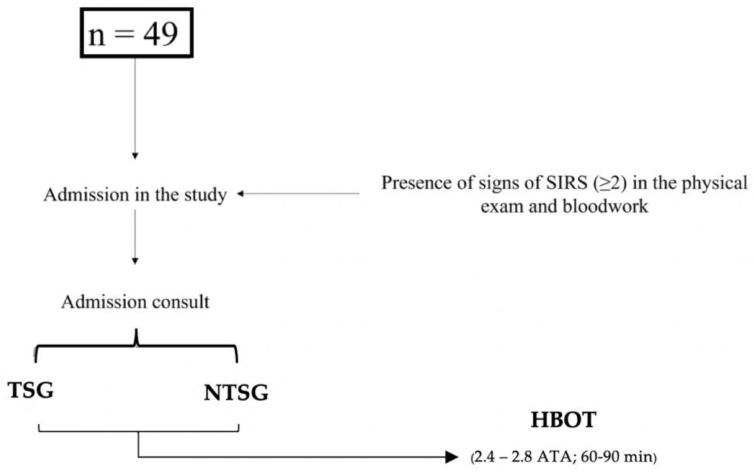
Diagram used for the admission of dogs in the clinical study. TSG, Traumatic Study Group; NTSG, Non-Traumatic Study Group; HBOT, Hyperbaric Oxygen Therapy.

**Figure 2 vetsci-09-00033-f002:**
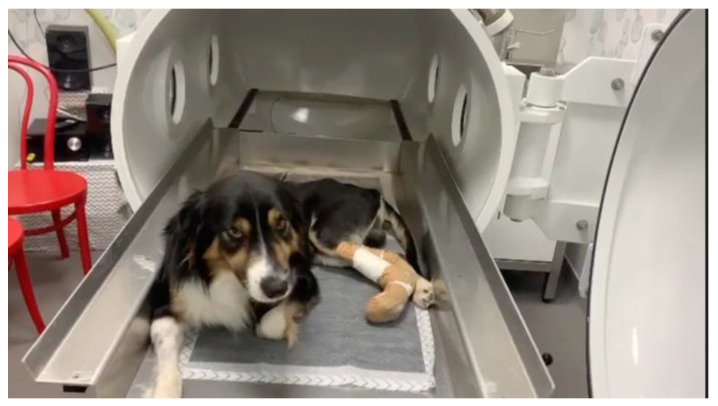
Dog placement inside the hyperbaric oxygen chamber.

**Table 1 vetsci-09-00033-t001:** Hyperbaric oxygen therapy main effects.

**Cytokines**	Increase of endothelin 1; vasoconstriction.
	Decrease of IL-1, IL-6 and TNF.
	Increase of VEGF; angiogenesis.
	Decrease of TNF-α after ischemia and reperfusion.
	Up-regulation of FGF.
**Prostaglandins**	Decrease of PGE2 in macrophages, bone, gingiva, colon, and kidney.
**Nitric Oxide**	Up-regulation of NO production.

Legend: IL-1, interleukin 1.; IL-6, interleukin 2; VEGF, vascular endothelial growth factor; TNF, tumor necrosis factor; FGF, fibroblast growth factor; PGE2, prostaglandin E2; NO, nitric oxide [5,6,7,8].

**Table 2 vetsci-09-00033-t002:** Systemic inflammatory response syndrome diagnostic parameters.

Heart Rate	>120 bpm
Respiratory Rate	>20 bpm
Temperature	<38 °C ou >39 °C
Leucogramme	<6000/µL ou >16,000/µL ou >3% band neutrophils

(Adapted from Hauptman et al., 1997 [9]).

**Table 3 vetsci-09-00033-t003:** Sample population characterization.

	%	Mean	Mode	Median	SD (+/−)	SEM
Breed	59.2% known breed 40.8% mixed breed	✗	✗	✗	✗	✗
Sex	55.1% male 44.9% female	✗	✗	✗	✗	✗
Etiology	65.3% TSG 34.7% NTSG	✗	✗	✗	✗	✗
Age	✗	8.845	8	9	4.8719	0.6960
Weight	✗	23.224	25	25	14.1705	2.0244

TSG, Traumatic Study Group; NTSG, Non-Traumatic Study Group; SD, standard deviation; SEM, standard error of the mean.

**Table 4 vetsci-09-00033-t004:** Hyperbaric oxygen therapy minor and major side effects.

Minor Side Effects	Major Side Effects
Head shaking	Barotrauma
Increase in respiratory frequency	Seizures
Yawning	Syncope
Ear scratching	Death
Swallowing	
Vocalizations	
Anxious after 5–10 min of treatment	

**Table 5 vetsci-09-00033-t005:** Parametric *t*-tests for independent samples and non-parametric tests.

	*t*-Test	Chi-Square	Mann-Whitney
*Etiology* and *number of HBOT sessions.*	✓	✓	✓
*Etiology* and *number of days between the first and the last HBOT sessions.*	✓	✓	✓
*Etiology* and *number of days between the diagnosis and the first HBOT session.*	✓	✓	✗
*Etiology* and *clinical outcome.*	✗	✓	✓

**Table 6 vetsci-09-00033-t006:** Monitorization of lactate, systolic arterial pressure, and modified Glasgow scale in TSG dogs.

Number of Dogs n = 32	Serum Lactate at Admission	Systolic Arterial Pressure at Admission	Modified Glasgow Scale at Admission	Monitorization
8	N	N	10 (n = 3) 9 (n = 5)	13–15 (n = 8 after 48 h)
4	N	N	N	✗
10	N	60–65 mmHg	N	N after fluid therapy resuscitation
7	>4 mg/dL	N	N	N Lact clearance (6 h of fluid therapy)
3	>10 mg/dL	N	N	Half of Lact clearence (6 h after fluid therapy), N Lact clearence (24 h)

TSG, Traumatic study group; N, normal; Lact, lactate.

**Table 7 vetsci-09-00033-t007:** Results of the side effects shown in the study population.

Side Effects	TSG (n = 32)	NTSG (n = 17)
Head shaking	32	17
Increase in respiratory frequency	21	17
Yawning	14	17
Ear scratching	14	12
Swallowing	0	1
Vocalizations	0	1
Anxious after 5–10 min of treatment	2	7

Legend: TSG, Traumatic Study Group; NTSG, Non-Traumatic Study Group.

## Data Availability

The data presented in this study are available upon request from the corresponding author.

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
