# Peer review of "Hyperbaric Oxygen Therapy in Systemic Inflammatory Response Syndrome"

_vetsci, 2022, doi:10.3390/vetsci9020033_

Round 1
Reviewer 1 Report
Extensive editing of the English language is required.
Author Response
Regarding the general comments, we thank the reviewer for all of the comments and every suggestion made. We think we could cope with all and that it will for sure improve and enrich this manuscript. Regarding the English language and style, we improved our english and resorted to one of the MDPI English Editing Services.
Reviewer 2 Report
The presented work evaluates the safety of hyperbaric oxygen therapy in the treatment of dogs with systemic inflammatory response syndrome, as well as its influence on the outcome. The study is relevant since, although there is ample evidence of the benefits of HBOT in mice, rats, and humans, limited clinical studies are available in dogs.
This is an interesting work and it is well written, but some sections could be improved.
The authors refer to the use of several antibiotics when in the presence of clinical signs of infection (section 2.3). However, it is not clear whether the antibiotic choice was confirmed/supported by culture and sensibility tests and if these were performed (always/never/in some cases). Considering the importance of sepsis as a cause for SIRS, the increasing numbers of antibiotic-resistant strains, and the responsible use of antibiotics, it would be relevant to clarify this point since it is relevant for the clinical outcome, too.
The authors refer in section 2.5 the use of the SIRS scale, qSOFA when suitable, and measurement of serum lactate and systemic arterial pressure, amongst others. However, the results of this evaluation were not presented in the results section.
The authors state that both parametric and non-parametric tests were used (Section 2.6) but it is not clear which ones for each pair of variables. A table should be included, detailing the applied tests in each case.
Considering there is no control group in the present study (understandable for ethical reasons since the aim is to provide the best possible care), in order to support the authors' conclusion indicating the efficacy of HBOT therapy in face of the present study results, it would be relevant to present the results of quantitative variables such as the ones referred in Section 2.5, possibly in the form of graphics showing the evolution of the parameters.
The safety of HBOT therapy in the described conditions is supported by the presented results.
Reviewer 3 Report
Gouveia and co-workers have assessed the safety, tolerability and applicability of HBOT in dogs positive for SIRS, due to traumatic or non-traumatic causes. The study is well designed and while of a relatively small cohort size (since the n=49 dogs had to be divided into two groups) they have made interesting and relevant findings, which will of no doubt be of interest to veterinary practitioners with canine patients positive for SIRS.
I only have minor suggested modifications:
Line 32: insert an “it” between “that” and “should”
Line 37: recommend changing “imply” to “evoke”
Table 1: references are needed
Lines 74-78: recommend changing to “We hypothesised that HBOT in positive SIRS dogs, of traumatic or non-traumatic origin, may be a useful adjuvant therapy. Thus, the aim of our study was to assess the safety, tolerability and applicability of HBOT in dogs positive for SIRS, due to traumatic or non-traumatic causes”.
Line 81: remove the comma between “prospective” and “cohort”
Line 82/83: change “addresses” to “addressed”
Line 100: this sentence is unnecessary and can be removed.
Figure 1: this is unnecessary as the admission procedure for t he dogs is already detailed in the text. Instead, consider replacing this figure with a table downing the details of the patient cohort as included in Table 1 (i.e., breed, sex, age, etc). Also, the age and weight should be shown as average +/- SD. Replace “mutts” with “mixed breed” or “unknown breed” (depending on which is appropriate).
Line 189: should read “The number of sessions….”
Table 4: the formatting needs fixing
Line 352: remove the first comma and the second occurrence of “that”
Line 382: remove the second comma
Line 395: insert a comma after “ATA”
Line 398: change to “not following that”
Line 399: suggest rewording to read “The treatments that are frequently used in human medicine in the resolution of chronic and difficult wounds are a pressure of 2.4 ATA, where the seizure incidence reported was as 1.3 occurrences in 10.000 treatments”
Line 401: should that read “10,000” treatments (not “10.000” treatments)?
Line 402: “Veterinary Medicine” (and also “Human Medicine”) does not need capitalization
Line 446: consider changing the last two sentences of the Discussion to come straight after the sentence ending “….and the clinical outcome” and say something like “Thus, future studies with larger cohort sizes are needed to further explore this paradigm”.
Line 454” change “interferes” with “affects”
Line 459: I am confused by the final sentence of the Conclusion. I don't feel it helps the authors in anyway and would suggest it be removed.
Line 458: Suggest removing the words “, supporting the study hypothesis”.
